# Shifted and Squeezed 8-bit Floating Point format for Low-Precision Training of Deep Neural Networks

**Léopold Cambier[1][*][†], Anahita Bhiwandiwalla[2][†], Ting Gong[2],**
**Mehran Nekuii[2], Oguz H Elibol[2] and Hanlin Tang[2]**
[1]ICME, Stanford University
[2]Intel AI Lab
lcambier@stanford.edu
{anahita.bhiwandiwalla,ting.gong}@intel.com
{mehran.nekuii,oguz.h.elibol,hanlin.tang}@intel.com

## Abstract

Training with larger number of parameters while keeping fast iterations is an increasingly adopted strategy and trend for developing better performing Deep Neural Network (DNN) models. This necessitates increased memory footprint and computational requirements for training. Here we introduce a novel methodology for training deep neural networks using 8-bit floating point (FP8) numbers. Reduced bit precision allows for a larger effective memory and increased computational speed. We name this method Shifted and Squeezed FP8 (S2FP8). We show that, unlike previous 8-bit precision training methods, the proposed method works out-of-the-box for representative models: ResNet-50, Transformer and NCF. The method can maintain model accuracy without requiring fine-tuning loss scaling parameters or keeping certain layers in single precision. We introduce two learnable statistics of the DNN tensors - *shifted* and *squeezed* factors that are used to optimally adjust the range of the tensors in 8-bits, thus minimizing the loss in information due to quantization.

## 1 Introduction

Deep neural networks have achieved state-of-the-art performance on a wide variety of computer vision, audio, and natural language processing (NLP) tasks. This has resulted in an explosion of interest around techniques to reduce the memory footprint and energy consumption of neural network training and inference (Guo, 2018). Although there are a number of methods to address some of these issues for inference, the most effective method for training is using reduced precision numerical formats.

While 32-bit floating point (FP32) is the most common data format for neural network training, recent hardware have leveraged techniques that allow for training with 16-bit data formats (Köster et al., 2017; Micikevicius et al., 2018). However, 8-bit precision training remains an open challenge (Johnson, 2018; Kalamkar et al., 2019). Current FP8 training methodologies (Wang et al., 2018; Mellempudi et al., 2019) require either specialized chunk-based accumulation, stochastic rounding techniques, loss scaling or maintaining some layers of the network in higher precision. Tuning these knobs is non-intuitive and requires significant experimentation for each individual network.

Accelerating the adoption of 8-bit data in training DNNs requires a hardware-friendly and out-of-the-box implementation of FP8. Due to the reduced number of mantissa bits, 8-bit multipliers are smaller and consume less power compared to higher bit representations. In this work we describe a novel 8-bit floating point (FP8) format - shifted and squeezed FP8 (S2FP8) - which has the following advantages compared to previously proposed 8-bit training methodologies:

---

[*]Work performed during an internship at Intel
[†]Equal contribution

- S2FP8 eliminates the need for loss scaling, which requires significant tuning of the loss scale values and schedule for individual topologies

- Leveraged by the forward and backward passes of model training, S2FP8 is effective in adjusting the range of gradients and also of activations and weights

- S2FP8 does not require keeping the first and last layer in FP32 precision, which is needed for other approaches (Mellempudi et al., 2019), however maintains the master weights and accumulations inside the matrix multipliers in FP32

We demonstrate across image classification, translation, and recommendation models that S2FP8 outperforms previous 8-bit approaches, and reaches the accuracy of FP32 models without any additional hyperparameter tuning.

## 2 RELATED WORK

The success of 32-bit floating point data type in training deep neural networks has increased interest in the feasibility of even lower precision training. The exponential demand for compute involved in training these deep neural networks has lead to multiple advancements in lower precision data types.

Several studies have developed techniques such as loss scaling, stochastic rounding, and others to train effectively in 16-bit (Micikevicius et al., 2018; Das et al., 2018; Azim), along with associated hardware support (Markidis et al., 2018). Using 16-bit fixed point, (Gupta et al., 2015) showed that stochastic rounding techniques were crucial for model convergence even for simple convolutional neural networks. As noted in (Kalamkar et al., 2019), Google's bfloat16 format has the same number of exponent bits as FP32, leading the success of that format without commonly requiring hardware intensive requirements such as stochastic rounding or other framework level techniques such as loss scaling.

Although 8-bit formats have significant performance and memory advantages, convergence is especially challenging due to loss of accuracy in the backpropogated gradient values. Wang et al. (2018) demonstrated training models with matrix multiplications and convolutions in FP8 but they use FP16 with chunk-based accumulations and stochastic rounding hardware. Mellempudi et al. (2019) also demonstrated success with FP8, accumulating in FP32 and using loss scaling techniques on ResNets, Transformer and GNMT networks. However, they too require the first and last layers of the model to be in FP32, and similar to (Banner et al., 2018) leverage Stochastic Rounding techniques to maintain model accuracy. Unlike S2FP8 proposed in this work, both of these FP8 training techniques emphasize the need for efficient loss scaling, rounding hardware and restriction on some layers being in higher precision.

Zhou et al. (2016) quantized weights, activations and gradients of AlexNet (Krizhevsky et al., 2012) to 1, 2 and 6 bits respectively. But they also need to maintain the first and last convolution layers in full precision and stochastically quantize the gradients. Wu et al. (2018) demonstrate using integers for training LeNet-5 (LeCun et al., 1998) and AlexNet with 8-bits for activations, error and gradients and 2-bits for weights. However, these approaches also required custom tuning such as novel initialization techniques and layer wise scaling instead of Batch Normalization and Softmax. These approaches lack generalizability to other models, requiring significant fine tuning.

To the best of our knowledge, there does not exist an out-of-the-box solution using FP8 in training deep learning topologies without the need for tuned loss scaling techniques, requirements of certain layers being in full precision along with efficient hardware rounding schemes like Stochastic Rounding.

## 3 SHIFTED AND SQUEEZED 8-BIT FLOATING POINT FORMAT

### 3.1 CHALLENGES OF 8-BIT FLOATING POINT FORMAT

The FP8 format, with 2 bits of mantissa and 5 bits of exponent (Mellempudi et al., 2019) is both narrow (i.e., its dynamic range is very limited, from $2^{-16}$ to $2^{16}$) and has lower accuracy (the machine epsilon is only $2^{-3}$). Figure A1 illustrates the range and accuracy of FP8. In contrast, FP32 ranges from $2^{-149}$ to $2^{128}$ with a machine-epsilon of $2^{-24}$ (Table A1).

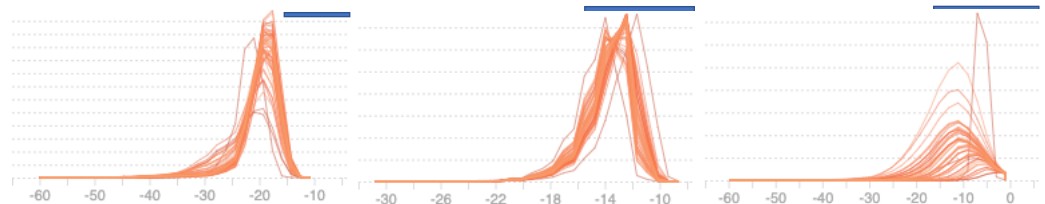

Figure 1: The distribution of tensor elements over the course of training for three tensors from the Transformer tiny model on the English-Vietnamese translation dataset. Blue bar indicates the representable range of FP8. Left: Many of the tensor elements fall outside of FP8's representable range. Center: Few tensor elements fall outside of FP8's representable range. Right: Initially, most elements are within FP8's representable range, but after training, many fall outside of the representable range

On the other hand, tensors involved in neural networks (weights, activations and gradients) are spread across varying scales. As illustrated in Figure 1, the tensor distributions change over the course of training, spanning different orders of magnitude.

As a result, 8-bit training usually requires a combination of multiple techniques to capture the full dynamic range of values for model training. Some of these techniques include:

- Loss scaling (Micikevicius et al., 2018) scales the loss $\mathcal{L}(w)$ by a constant $\lambda$ before back-propagation . This makes the gradients artificially larger, allowing them to fit within the FP8 range. Gradients are then scaled down before being accumulated into the trainable weights as shown in Equation 6

- Stochastic rounding (Maxfield, 2006) alleviate quantization errors by capturing some of the information discarded when truncating to lower precision at the output of a GEMM operation

Between these two techniques, loss scaling is more critical; once the magnitude of the gradients can no longer be represented in the FP8 range, training convergence will not be possible. However, loss scaling only modifies the gradients. Weights and activations can also (albeit admittedly less frequently) exceed the FP8's representable range of $[2^{-16}, 2^{16}]$. In those scenarios, convergence can also be affected.

The issue with loss scaling is that it requires user interaction. Models have to be modified, and, more importantly, tedious empirical tuning is required to find the correct loss scaling schedule. While some networks can be trained with constant loss scaling, some, notably Transformers (Mellempudi et al., 2019), require dynamic "back-off" and improved loss scaling. This requires significant trial and error to tune the scaling schedule, slowing down wide adoption of low-precision numerical formats.

## 3.2 Shifted and Squeezed FP8

To alleviate these issues and make neural network training possible with no model modifications or hyperparameter tuning, we propose a new 8-bit floating point format. Consider a tensor $X$ of size $N$, i.e., $X = \{X_i\}_{i=1}^N$. Instead of directly encoding each $X_i$ in FP8, we store $X$ using $N$ FP8 numbers $\{Y_i\}_{i=1}^N$ accompanied by two (squeeze and shift) factors $\alpha$ and $\beta$ (the "statistics" — see Figure 2).

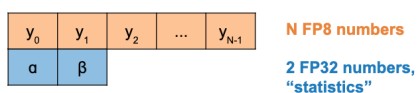

Figure 2: The S2FP8 format. A tensor $X$ of $N$ numbers is represented by $\alpha$, $\beta$ and $N$ FP8 numbers $Y$, related to $X$ through Equation 1.

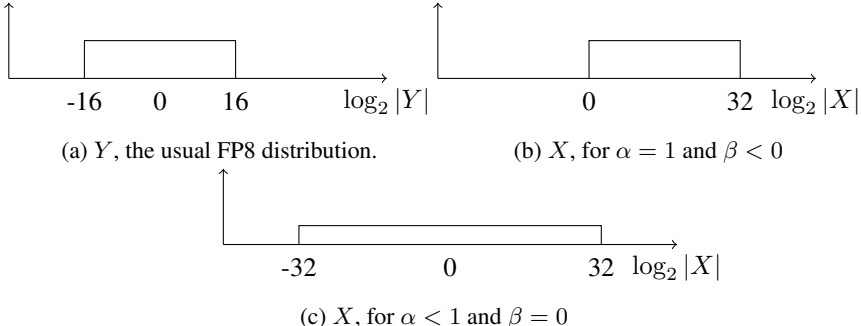

(a) $Y$, the usual FP8 distribution.

(b) $X$, for $\alpha = 1$ and $\beta < 0$

(c) $X$, for $\alpha < 1$ and $\beta = 0$

Figure 3: Impact of the Shifted and Squeezed transformation $\log_2 |Y| = \alpha \log_2 |X| + \beta$. $\alpha$ let the distribution be as wide as necessary (though, with an associated loss of precision), and $\beta$ let us shift the distribution around any value.

For $X_i \neq 0$, $X$ and $Y$ are then related through

$$\log_2(|Y_i|) = \alpha \log_2(|X_i|) + \beta \Leftrightarrow Y_i = \pm 2^\beta |X_i|^\alpha \tag{1}$$

where the $\pm$ is chosen so that $X_i$ and $Y_i$ have the same sign. This representation allows for $\alpha$ and $\beta$ be chosen so that together with tensor $Y$ they capture most of the dynamic range of the tensor $X$. As we will see in section 4, this is all that is necessary to train networks using 8-bit floating point numbers.

In order for $Y$ to be a tensor suitable to be represented by FP8 numbers, we enforce that it has zero mean and a maximum value within the dynamic range of FP8 (e.g. 15):

$$\sum_{i=1}^{N'} \log_2(|Y_i|) = 0 \quad \text{and} \quad \max_{i=1,\dots,N'} \log_2(|Y_i|) = 15(= \log_2(2^{15})) \tag{2}$$

where the $'$ notation indicates that the sum and the max, respectively, ignore any $i$ such that $Y_i = 0$. Those equations ensure that $\log_2(|Y|)$ values are distributed with zero mean and each is less than 15, which is ideal for an FP8 format.

By inserting Equation 2 into Equation 1, and by denoting

$$\mu = \sum_{i=1}^{N'} \log_2(|X_i|) \quad \text{and} \quad m = \max_i \log_2(|X_i|) \tag{3}$$

we find

$$\alpha = \frac{15}{m - \mu}, \qquad \beta = -\alpha\mu \tag{4}$$

This new tensor format results in the training procedure (forward pass, backward pass, weight update) described in Figure 4. Forward and backward MatMul use this new S2FP8 format. Master weights are kept in FP32 and updated using S2FP8 gradients. Accumulations inside the GEMM kernel are kept in full FP32 precision. Figure 3 illustrates the impact of $\alpha$ and $\beta$. By having those two extra degrees of freedom for each tensor, majority of the dynamic range of each tensor can now be captured, whether very small ($\beta > 0$), very large ($\beta < 1$), very narrow ($\alpha > 1$)) or very wide ($\alpha < 1$).

### 3.3 LEARNING THE TENSOR DISTRIBUTION

One way to interpret $\alpha$ and $\beta$ is to consider them as parameters of a distribution generating the tensor values $\log_2(|X_i|)$. We can then say that, by continuously computing $\alpha$ and $\beta$, we are effectively learning the distribution of $\log_2(|X_i|)$. Figure 5c shows the evolution of $\mu$, $m$, $\alpha$ and $\beta$ for a particular tensor of ResNet-20. We see that $\alpha$ and $\beta$ converge to, approximately, 5 and 21, respectively. From Equation 1, we conclude that:

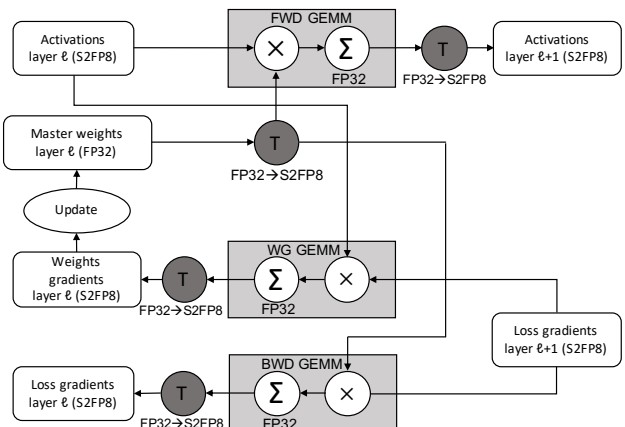

Figure 4: Low precision training with S2FP8. $T$ represent the truncation described in Equation 5, from FP32 to S2FP8. When using S2FP8 for training, forward and backward GEMM's only use S2FP8. The master weights are kept in FP32 and updated during the update step.

- since $\alpha > 1$, this means that $X$ is *expanded* into $Y$, i.e., $X$ is more *narrow* than what FP8 allows
- since $\beta > 0$, this means that $X$ is *right-shifted* into $Y$, i.e., $X$ is *smaller* than what FP8 allows

At convergence, those $\alpha$ and $\beta$ values represent the distribution of each converged tensor. Notice that all statistics stabilize in the last third of the training, where the learning rate is decreased, indicating the network is converging to its final state.

## 4    EXPERIMENTAL RESULTS

In this section, we compare S2FP8 training with baseline FP32 and FP8 training with and without loss scaling for: Residual Networks (He et al., 2016) of varying depths on the CIFAR-10 and ImageNet (Deng et al., 2009) datasets, Transformer (Vaswani et al., 2017) on IWSLT'15 English-Vietnamese dataset (Luong & Manning, 2015), and Neural Collaborative Filtering (NCF) (He et al., 2017) on MovieLens 1 Million dataset (Harper & Konstan, 2016).

For our experiments, we use the open source Tensorflow Models[1] repository for ResNet and NCF, Tensor2Tensor (Vaswani et al., 2018) for Transformer with added S2FP8 data type simulation support using the methodology described in subsection 4.1. For a given model, we keep the hyperparameters consistent across FP32, FP8 and S2FP8 evaluations.

### 4.1    SIMULATION METHODOLOGY

We simulated S2FP8 by inserting appropriate truncation function throughout the network, before and after every convolution and matrix-matrix product operations, during both the forward and backward passes. The rest of the network is kept in FP32, and those truncation simulate the low-precision training described in subsection 3.2.

The truncation function takes as input a tensor $X$, computes its magnitude mean and maximum, computes the appropriate $\alpha$ and $\beta$ and finally truncates $X$ by computing

$$X_{truncated} = \left[2^{-\beta} \left\{\text{truncate}_{\text{FP8}}\left(2^{\beta}|X|^{\alpha}\right)\right\}\right]^{1/\alpha} \tag{5}$$

where $\text{truncate}_{\text{FP8}}$ is a usual FP8 truncation function with RNE (round-to-nearest, with ties broken by rounding to even) rounding which is easier to implement and most widely supported in hardware.

---

[1]https://github.com/tensorflow/models

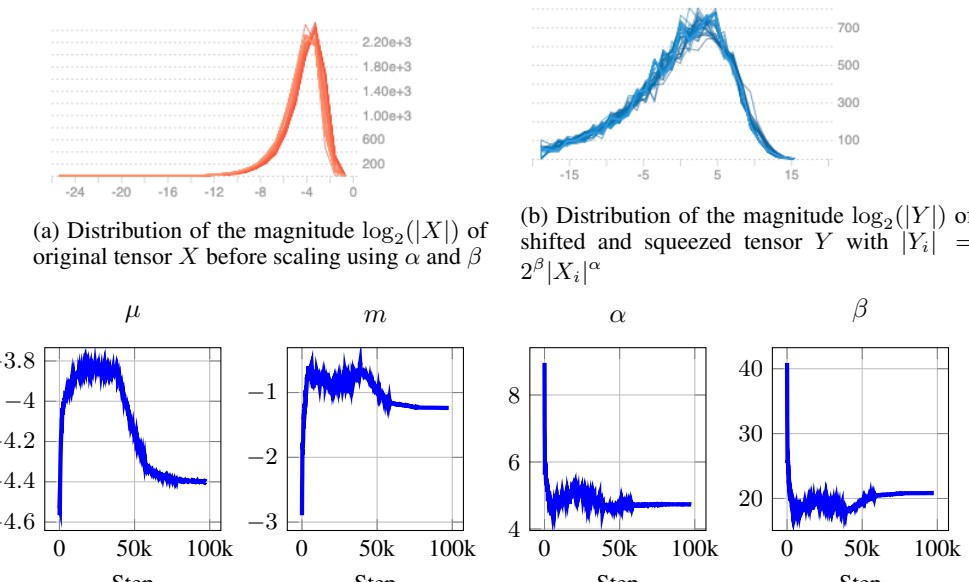

(a) Distribution of the magnitude $\log_2(|X|)$ of original tensor $X$ before scaling using $\alpha$ and $\beta$

(b) Distribution of the magnitude $\log_2(|Y|)$ of shifted and squeezed tensor $Y$ with $|Y_i| = 2^\beta |X_i|^\alpha$

(c) The computed statistics during training for the scale ($\beta$), shift ($\alpha$), as well as the mean of the log values ($\mu$) and the maximum log value ($m$).

Figure 5: Evolution of the average and maximum magnitude, as well as $\alpha$ and $\beta$ for CIFAR-10 with ResNet-20. This illustrates how the network is actually implicitly learning the tensors distribution, by repeatedly computing magnitudes $\alpha$ and $\beta$ through $\mu$ and $m$.

## 4.2 RESIDUAL NETWORKS

We first present results with Residual Networks of varying depths on the CIFAR-10 image recognition dataset. We trained the model on 1 GPU using standard parameters: 250 epochs, batchsize of 128, SGD with momentum of 0.9, initial learning rate of 0.1 decreased by a factor of 10 after epochs 100, 150 and 200.

Table 1 and Figure A2 presents the results. We observe that S2FP8 reaches almost exactly the FP32 baseline, sometimes even improving over it. Out-of-the-box FP8 does not converge and has very poor accuracy. Finally, FP8 with constant loss scaling of 100 (FP8+LS(100)) can reach the baseline. Both S2FP8 and FP8+LS(100) have similar performances, but S2FP8 can do so without any extra hyperparameters or tuning from the user's perspective.

| CIFAR-10 | FP32 | S2FP8 | Δ | FP8 | FP8+LS(100) |
|---|---|---|---|---|---|
| ResNet-20 | 91.5 | 91.1 | 0.4 | 17.9 | 91.1 |
| ResNet-34 | 92.5 | 92.0 | 0.5 | 13.5 | 92.0 |
| ResNet-50 | 93.0 | 93.2 | -0.2 | 11.5 | 92.9 |

Table 1: Validation accuracy (in %) for image recognition on CIFAR-10 with ResNet-20/34/50.

We also evaluate S2FP8 on the 1000 class ImageNet dataset. Here, we trained the network on 4 GPUs using standard parameters: 90 epochs, batchsize of 256, SGD with momentum of 0.9, initial learning rate of 0.1 decreased by a factor of 10 after epochs 30, 60, 80 and 90. Table 2 and Figure 6 present the results.

Again, we observe that S2FP8 gets very close to the FP32 baseline. Out-of-the-box FP8 quickly diverges and does not converge at all. For FP8 with loss scaling to converge, one has to not truncate the first and last layer, as consistent with (Mellempudi et al., 2019), which we denote as Ex in Table 2 below. A loss scaling of 10,000 can then be used to reach the baseline (FP8+LS(10k)+Ex). Finally, stochastic rounding can be added and it slightly improves the precision (FP8+LS(100k)+Ex+SR). However, both those cases are not out-of-the-box, as they require loss scaling tuning and some layers

to be kept in full precision. S2FP8 does not suffer from that, thanks to its improved quantization: all layers can be truncated and no loss scaling is required.

| Imagenet1k | FP32 | S2FP8 | Δ | FP8 | FP8+LS(10k)+Ex | FP8+LS(100k)+Ex+SR |
|---|---|---|---|---|---|---|
| ResNet-18 | 70.3 | 69.6 | -0.7 | NaN | 68.7 | 68.9 |
| ResNet-50 | 76.2 | 75.2 | -1.0 | NaN | 75.3 | 75.5 |

Table 2: Validation accuracy (in %) for image recognition on Imagenet1k with ResNet-18/50

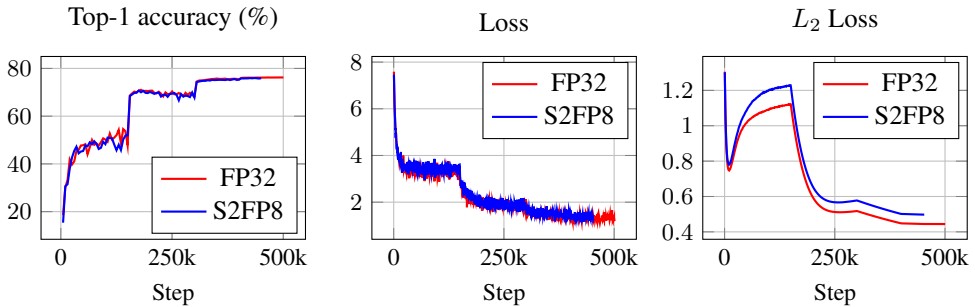

Figure 6: Comparing Top-1 accuracy and Loss of S2FP8 with FP32 for ResNet-50 on Imagenet1k

### 4.3 TRANSFORMER

We also tested S2FP8 on a small Transformer (Transformer Tiny) on the English-Vietnamese dataset. The model has 2 hidden layers of size 128, and a filter of size 512, and is trained using Adam optimizer (Kingma & Ba, 2014).

Table 3 and Figure 7 show the result, where we compare FP32, S2FP8 and FP8 with exponential loss scaling. We tried many loss scaling schedules (constant and exponential, with various initializations) and report the best result. As one can see, S2FP8 reaches the baseline with no hyperparameter tuning. FP8, on the other hand, does not, even after extensive loss scaling tuning. This shows the value of an out-of-the-box method for the user.

| En-Vi | FP32 | S2FP8 | Δ | FP8 | FP8+LS(exp) |
|---|---|---|---|---|---|
| Transformer tiny | 25.3 | 25.3 | 0.0 | NaN | 21.3 |

Table 3: BLEU Score (Papineni et al., 2002) (from 0 to 100) for translation task on the English-Vietnamese dataset with Transformer tiny.

### 4.4 NEURAL COLLABORATIVE FILTERING

The Neural Collaborative Filtering (NCF) network comprises of embeddings for users and items from the MovieLens dataset, that are then passed to a Multi-Layer Perceptron(MLP) network to learn the user-item interaction. Matrix-multiplication operations are the building blocks of such models. We compare S2FP8 with FP32 and FP8 without loss scaling. We simulate Matrix-Multiplications and look-ups from the embeddings in S2FP8 and compare it to FP8 with RNE. We trained the model on the MovieLens 1 Million dataset with the following standard paramaters: 20 iterations, batchsize of 1024 on 4 GPUs, 8 predictive factors, learning rate of 0.0005 using the Adam optimizer. Figure 8 and Table 4 show the result, where we compare FP32, S2FP8 and FP8 without loss scaling.

This again shows that S2FP8 easily reaches the baseline out-of-the-box, without tuning of any sort. FP8 gets relatively close, but cannot reach the baseline.

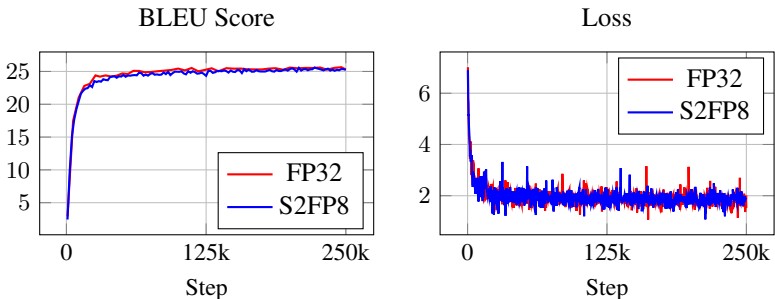

Figure 7: Comparing BLEU score and Loss of S2FP8 and FP32 for Transformer tiny on En-Vi dataset

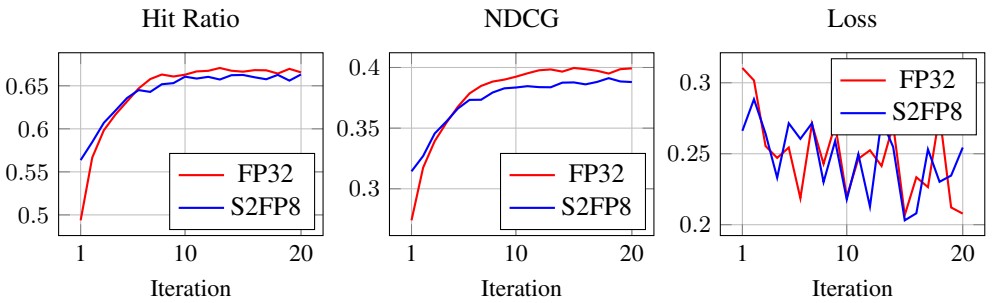

Figure 8: Comparing Hit Ratio, NDCG and Loss of S2FP8 and FP32 for NCF on MovieLens-1M

## 5 HARDWARE ASPECTS

S2FP8 is a new data type and requires its own circuitry to be implemented in a tensor processing engine. However, the added overhead is very minimal and affects neither data throughput nor compute speed. In order to convert FP32 tensors into S2FP8, two hardware (HW) components are needed. One is to calculate each tensor's statistics (Equation 3), which bring minimal HW complexity. To make compute operations even easier these statistics could be stored in lower precision such as FP8/INT8. The other component is to adjust the exponent and mantissa of all those tensor elements by applying the squeeze ($\alpha$) and shift ($\beta$) factors in Equation 4 before truncating them into their 8-bit placeholders. The shift could be done using simple element-wise add/subtract operations on the exponents, and element-wise squeeze could be applied to the mantissa portions. Another consideration is within the tensor processing engine(e.g., GEMM engine) which requires the $\alpha$ and $\beta$ factors while doing the calculations. The FP32 result will be converted back to S2FP8 when needed (e.g., to store back in memory) as shown in Figure 4.

## 6 CONCLUSION

We introduce a novel 8-bit floating point data type (S2FP8), that gives competitive performance in comparison to state-of-the-art FP32 baselines over a range of representative networks. S2FP8 makes use of *shifted* and *squeezed* factors to shift and rescale the range of tensors prior to truncation. S2FP8 allows training of neural networks with an 8-bit format while eliminating the need for loss scaling tuning, hardware-complex rounding techniques. In addition, compared to existing FP8 implementations we also eliminate the restriction of maintaining the first and last layers in FP32. Decreasing

| Movielens 1 million | FP32 | S2FP8 | $\Delta$ | FP8 |
|---|---|---|---|---|
| NCF | 0.666 | 0.663 | 0.003 | 0.633 |

Table 4: HR Score for NCF on the Movielens 1 million dataset.

the number of bits enables larger models to fit on a single device and results in faster training. As part of future work, we plan to extend the use of S2FP8 to train additional DNN topologies and also simplify the *squeeze* and *shift* statistics from a hardware implementation point of view. We also plan to explore the use of reduced precision to store the statistics and the extendability of this approach to efficiently represent a broader suite of low precision formats like 8-bit POSIT (Gustafson & Yonemoto, 2017), 4-bit floating and integer data types.

ACKNOWLEDGMENTS

We would like to thank Naveen Mellempudi, Pratap Prasad, Prasanna Singamsetty and Cory Stephenson for insightful discussions.

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

# A   APPENDIX

## A.1   SUPPLEMENTARY TABLES AND FIGURES

| Format | Bits | s/e/m | Min sub-normal | Min nor-mal | (Approx.) Max normal | Machine epsilon | Range |
|--------|------|-------|----------------|-------------|----------------------|-----------------|-------|
| IEEE-FP32 | 32 | 1/8/23 | $2^{-149}$ | $2^{-126}$ | $2^{128}$ | $2^{-24}$ | $2^{277}$ |
| IEEE-FP16 | 16 | 1/5/10 | $2^{-24}$ | $2^{-14}$ | $2^{16}$ | $2^{-11}$ | $2^{40}$ |
| BF16 | 16 | 1/8/7 | $2^{-133}$ | $2^{-126}$ | $2^{128}$ | $2^{-8}$ | $2^{261}$ |
| FP8 | 8 | 1/5/2 | $2^{-16}$ | $2^{-14}$ | $2^{16}$ | $2^{-3}$ | $2^{32}$ |

Table A1: Comparing several floating point formats. s/e/m indicates the number of sign (s), exponent (e) and mantissa (m) bits.

| Models | Datasets | FP32 | BF16 | FP8 | FP8+other recipes | S2FP8 |
|--------|----------|------|------|-----|-------------------|-------|
| ResNet-20 | CIFAR-10 | 91.5 | 91.7 | 17.9 | 91.1(Loss Scale=100) | 91.1 |
| ResNet-50 | CIFAR-10 | 93.0 | 93.2 | 11.5 | 92.9(Loss Scale=100) | 93.2 |
| ResNet-50 | ImageNet | 76.2 | 76.5 | NaN | 75.3(Loss Scale=10K, FP32 for first and last layers) | 75.2 |
| NCF | MovieLens1M | 0.666 | 0.653 | 0.633 | - | 0.663 |
| Transformer-tiny | En-Vi | 25.3 | 25.6 | NaN | 21.3(Loss Scale=Exp) | 25.3 |

Table A2: Comparing FP32, BF16, vanilla FP8, FP8 with tuning and S2FP8 on the model ResNet(Top1-accuracy), NCF(Hit Ratio),Transformer-tiny(BLEU score).

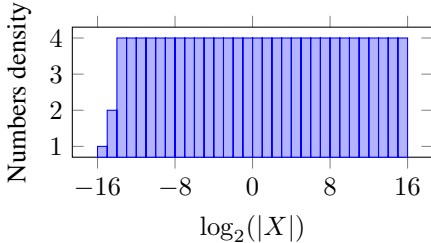

Figure A1: The range and precision of FP8. Bar indicate the number density between each power of 2. Since FP8 has 2 mantissa bit, the density is 4 (except in the denormals), and the associated machine epsilon is $2^{-3} = 1/8$. The normal representable range goes from $2^{-14}$ to $(1 - 2^{-3})2^{16}$, with denormals from $2^{-16}$ to $2^{-14}$.

## A.2   SUPPLEMENTARY EQUATIONS

$$\frac{\partial(\lambda\mathcal{L})}{\partial w}(w) = \lambda\frac{\partial\mathcal{L}}{\partial w}(w) \Rightarrow w^{(k+1)} = w^{(k)} - \alpha\frac{1}{\lambda}\frac{\partial(\lambda\mathcal{L})}{\partial w}(w^{(k)}). \tag{6}$$

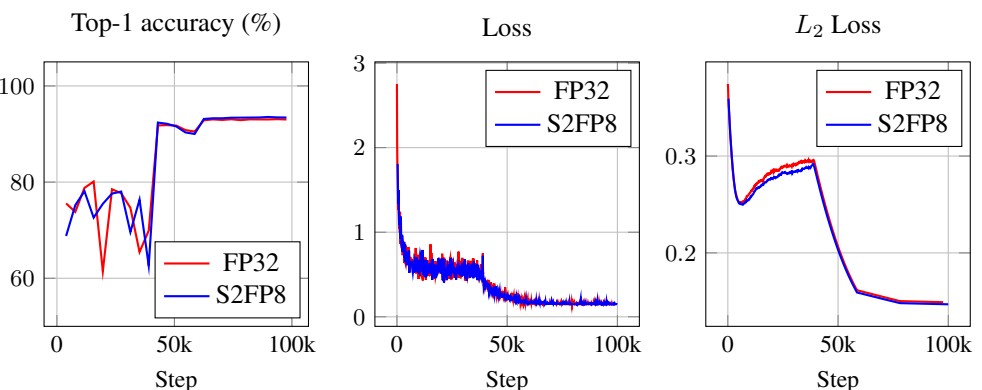

Figure A2: Convergence of ResNet-50 with the CIFAR-10 dataset

