# OpenReview forum: "Shifted and Squeezed 8-bit Floating Point format for Low-Precision Training of Deep Neural Networks"
_ICLR.cc/2020/Conference — Accept (Poster)_

### Official Review · AnonReviewer2 · 2019-10-23
**Official Blind Review #2**

**Rating:** 6

**Review:**

There has been a great deal of interest and research into reduced numerical precision of weights, activations and gradients of neural networks. If, for example, 16 bit floating point can be used instead of 32 bit floating point, then the memory bandwidth is halved along with significant gains in computational performance.

In this work the authors propose an 8-bit floating point format (denoted S2FP8) for tensors. In general, computing activations and gradients with such low precision at training time, has generally proved challenging without a number of tricks such as scaling the loss of each minibatch to within a reasonable range. Such ``tricks'' can be difficult to tune for each problem.

The key idea here is that for each tensor of 8-bit numbers, two 32 bit floating point statistics are recorded as well. These determine (in log-space) a scale and an offset for the 8-bit numbers (eq 1). This means that in this format tensors of significantly different scales can be well-represented (although larger scales necessarily implies low precision).

Matrix multiplications are done in FP32 precision and then converted in S2FP8 format. This requires an additional step to accumulate the summary statistics of each tensor in order to convert from FP32 to S2FP8 (the mean and the max of the tensor elements in log space).

The weights of the network are stored in FP32 and the gradients and activations are computed in S2FP8 and used to update the weights.

They test this approach in ResNet, a small transformer and a MLP for collaborative filtering. They find it reaches similar performance to FP32 where standard FP8 format has worse performance or results in Nan's.

Improvements in computational efficiency, both at training and inference, are active areas of research and this work contributes a novel approach using summary statistics. However, there are a several ways this work could be improved.

1. There is no comparisons with bfloat16, which is becoming a widely used approach to lower precision and is gaining significant hardware support [1].

2. Discussion and analysis regarding the need to gather summary statistics after each matrix multiplication (or other tensor operation). It is claimed that this brings minimal HW complexity, but this doesn't seem well justified. For a large tensor, this additional reduction to compute statistics may be expensive (in memory bandwidth and computation), particularly since this is done with FP32.

3. Even with the current implementation on a GPU, it should be possible with kernel fusions to gain some significant savings in memory bandwidth (and therefore computational speed), but there is no attempt anywhere to show any runtime benefit on current hardware.

Minor issues:

Some captions are very terse and the figures would benefit from a clearer explanation (e.g. figure 6).

[1] https://en.wikipedia.org/wiki/Bfloat16_floating-point_format

**Experience Assessment:**

I have read many papers in this area.

**Review Assessment: Checking Correctness Of Derivations And Theory:**

I assessed the sensibility of the derivations and theory.

**Review Assessment: Checking Correctness Of Experiments:**

I assessed the sensibility of the experiments.

**Review Assessment: Thoroughness In Paper Reading:**

I read the paper at least twice and used my best judgement in assessing the paper.

---

> ### Author Response · Authors · 2019-11-12
> **Response to Reviewer 2**
>
> We thank the reviewer for their detailed evaluation and valuable feedback. Below are detailed responses to each suggestion:
>
> Q1.
> Please see the below table comparing FP32, BF16, vanilla FP8, FP8 with tuning and S2FP8 on the models already published in the paper. We have also updated the paper by adding this table to the Appendix(Table A2). Note that here a “BF8” would be equivalent to FP8.
>
> —————————————————————————————————————————————
> Model            |  Dataset           |  FP32    |  BF16   |  FP8      |  FP8+other recipes           |  S2FP8     |
> —————————————————————————————————————————————
> ResNet-20     |  CIFAR-10         |  91.5     |  91.7    |  17.9     |  91.1(Loss Scale=100)       |  91.1        |
> —————————————————————————————————————————————
> ResNet-50     |  CIFAR-10         |  93.0     |  93.2    |  11.5     |  92.9(Loss Scale=100)       |  93.2        |
> —————————————————————————————————————————————
> ResNet-50     |  ImageNet       |  76.2     |  76.5    |  NaN    |  75.3(Loss Scale=10K,        |  75.2        |
> 		       |		             |		     |	             |              | FP32 for First and Last     |                  |
> 		       |			     |		     |		     |	             |   Layers) 		             |                  |
> —————————————————————————————————————————————
> NCF                |  MovieLens1M |  0.666  |  0.653  |  0.633   |                -                             |  0.663      |
> —————————————————————————————————————————————
> Transformer |  En-Vi                |  25.3     |  25.6    |   NaN    |   21.3                                  |  25.3         |
> (tiny)              |			     |		     |              |               |(Loss Scale=Exponential)|                  |
> —————————————————————————————————————————————
> Table: Comparing FP32, BF16, vanilla FP8, FP8 with tuning and S2FP8 on ResNet(Top1-accuracy), NCF(Hit Ratio), Transformer-tiny(BLEU score)
>
> Q2.
> We agree with the reviewer that statistics computations used in this method would result in added complexity. However, the total complexity is negligible compared to implementing full 16-bit circuitry. For further details we refer the reviewer to the detailed comment we provided for Reviewer #1. In addition, the summary statistics can be in-lined with the streaming of data from the compute element to the memory, thus saving on memory bandwidth.
>
> Q3.
> Certainly fusion would help. Commonly ReLU like operations are fused with GEMM operations. But with fusion, the memory write is only moved to after the ReLU instead of the GEMM operation. Memory bandwidth still remains a bottleneck. Using fusion, our approach would further save memory, increase throughput and improve the speed of the compute engine. Our simulator is currently set up to test convergence accuracy only.
>
> Minor Suggestion -  Some captions are very terse and the figures would benefit from a clearer explanation   (e.g. figure 6).
> Thank you for the suggestion. We have updated the captions of  Figures(6,7,8) to make them clearer.

---

### Official Review · AnonReviewer1 · 2019-11-04
**Official Blind Review #1**

**Rating:** 1

**Review:**

The paper suggest the method to train neural networks using 8 bit floating point precision values. The method requires a hardware modification to evaluate per tensor statistics, which intuitively capture dynamic range that a given tensor values may fall into. Those per tensor statistics are used to map 8-bit representation into a rational number. Statistics themselves are calculated using 32 bit floating point numbers.

While the method is interesting, I do not think it is practical due to the required hardware modifications. I am by no means not a hardware design expert, but I am not convinced that the gain of using 8 vs 16 bit floating point numbers outweights any extra complexity of hardware implementation. Mapping between representation and actual numerical values is much more complex than when using standard floating point representation (as any given numerical value is defined by a single its 8 bit representation + alpha + beta tensor statistics), integrated circuitry used to execute arithmetic operations would likely be much more complex than when using standard floating point operations. It is plausible that the added complexity (in terms of transistor count) would negate any potential price or energy savings over simply using 16 bit floating point representation. This should definitely be discussed in depth in this paper where the main contribution is an algorithm to be implemented in hardware.

As different layers will have different (alpha,beta) tensor statistics, from the computational perspective they would look like different data types which would be cast into the same common type before doing any arithmetic operations on them. This would almost certainly require extra computational steps to be done potentially negating computational benefits over simply using 16 bit floating point numbers.

The algorithm evaluates per-tensor and not per-tensor-element statistics. This may work well in cases where all entries in a given tensor are of the same dynamic range, but may break down in other cases: for example, diagonal entries in a matrix multiplication layer inside LSTM may have very different statistics comparing to non diagonal entries.


**Experience Assessment:**

I do not know much about this area.

**Review Assessment: Checking Correctness Of Derivations And Theory:**

I assessed the sensibility of the derivations and theory.

**Review Assessment: Checking Correctness Of Experiments:**

I assessed the sensibility of the experiments.

**Review Assessment: Thoroughness In Paper Reading:**

I read the paper at least twice and used my best judgement in assessing the paper.

---

> ### Author Response · Authors · 2019-11-12
> **Response to Reviewer 1**
>
> Thanks to the reviewer for the detailed comments regarding hardware implementation. As pointed out by Reviewer #2, the bandwidth savings of moving to lower precision significantly improve both performance and power efficiency. Moving bytes is expensive compared to the relatively few additional FLOPS required for the quantization operation to S2FP8. This saving on memory and bandwidth is the main benefit of our proposed scheme.
>
> Furthermore, to quantify the computational complexity difference, we have to consider two scenarios. First is when we have the same squeeze factor for the two operands, and the other scenario is when they are different.
>
> In the first scenario, compared to FP16, S2FP8 requires fewer FLOPS to execute the same GEMM, even accounting for the conversion cost. To quantify, let’s assume as a baseline we are comparing with a GEMM engine which can multiply two tensors of size NxN containing FP16 elements. The operation needs N^3 flops (FP16 ops) and generates FP32 numbers to be truncated to FP16. Now assume using our custom-build GEMM engine. The inputs are in S2FP8 - the GEMM engine operates on these tensors considering the alpha and beta factors of each input tensor to obtain the final FP32 results. Since the elements are 8 bits, the approximate flops (in FP16 scale) would be ⅛*N^3. Next step is to convert this FP32 tensor to S2FP8. The statistics calculation operations and the scaling ops are element-wise operations which require approximately 2*4*N^2 flops (in FP16 scale). So the total flops changes from N^3 to ⅛*N^3+8*N^2. In the case of 128x128 tensors this results in 5x fewer FLOPS comparing S2FP8 with FP16.
>
> In the second scenario (where the operand’s squeeze factors are different), a naive approach would be to expand the two operands to 16 bit tensors before performing GEMM operation. Compare to a FP16 GEMM engine, this would add some additional FLOPS (10% as an example for 128x128 tensors). However, since the tensors are still in 8 bit (plus some statistics), some hardware design for specialized multipliers is possible to realize the FLOP savings above which are well worth the die area on chip.
>
> Thus as a whole, with the proposed method, the compute complexity would be less compared to a 16-bit GEMM engine.  At the same time, it would bring along 2x memory and 2x bandwidth savings.
>
> The reviewer’s point about the tensor’s dynamic range is a good point. We have not encountered those situations yet, but could employ several approaches. For example, in convolutional networks we are considering performing independent conversions per each convolution channel. In that case, if the tensors have different statistics on different convolution channels, then we can perform S2FP8 conversion on each individual channel. The increase in memory is relatively negligible. With tensor-wide S2FP8 conversion, a tensor takes N*C*H*W + 2*4 bytes to be stored (where N is the batch size, C is the number of input channels and H and W are height and width of the input image). At the channel level it requires N*C*H*W + C*2*4 bytes. In most of the cases N*H*W >> 8 which means negligible increase in memory requirements.
>
> An exact hardware implementation is out of scope for this paper, and we believe the exact implementation should not be necessary for acceptance. We’ve demonstrated the desirable properties of this numeric, and provided reasons above as to why it would bring compute and power efficiency gains compared to FP16, even with the added conversion operations. Please let us know if you have further questions or concerns.

---

### Official Review · AnonReviewer4 · 2019-11-05
**Official Blind Review #4**

**Rating:** 8

**Review:**

The paper focuses on training neural networks using 8-bit floating-point numbers (FP8). The goal is highly motivated: training neural networks faster, with smaller memory footprint and energy consumption.
The proposed approach S2FP8 eliminates the need for loss scaling and does not require keeping some of the layers in FP32 precision as in Mellempudi et al. (2019).
The paper is well written, easy to follow, and provides a detailed background for readers who are not knowledgeable in this field.

On the downside, the first sections give the impression that FP32 is not used at all: "S2FP8 does not require keeping the first and last layer in FP32 precision, which is needed for other approaches (Mellempudi et al., 2019).". However, Section 3.2 says that "Master weights are kept in FP32" and that "Accumulations inside the GEMM kernel are kept in full FP32 precision". I think this should be stated earlier, because otherwise, the introduction is overclaiming.

The evaluation is very convincing - the approach is demonstrated for image classification, Transformer-based translation, and neural collaborative filtering. S2FP8 outperforms previous FP8 approaches and reaches the accuracy of FP32 out-of-the-box.
It would be interesting to see the Transformer results with S2FP8 on additional datasets rather than the English-Vietnamese dataset only, which is considered a low-resource dataset, and on Transformer-base rather than "tiny". If you have such results on additional NMT datasets, I would be interested to see them, even if the performance of S2FP8 is worse than FP32 (I will not reduce my rating because of lower results there).

Overall, this is a good paper that presents an important contribution. Thus, I recommend accepting this paper.

Minor:
* Section 2, paragraph 3: "(Mellempudi et al. 2019) also demonstrated .." the name of the authors should be outside of the parentheses (\citet). Same in 4th paragraph: "(Zhou et al., 2016) quantized .."
* Equation (2) is a little difficult to read because of the sequence "and max log", in which each word has a different role. It might worth to break it into two lines, add brackets, or use logical "and" instead of the word "and"?
* Equation (3) uses "i-prime" in the argument for "max", but "i-prime" is not used.
* Figure 4 is referenced before Figure 3, this is a little confusing (the reader needs to scroll down for Figure 4, and scroll up for Figure 3).



**Experience Assessment:**

I do not know much about this area.

**Review Assessment: Checking Correctness Of Derivations And Theory:**

I carefully checked the derivations and theory.

**Review Assessment: Checking Correctness Of Experiments:**

I carefully checked the experiments.

**Review Assessment: Thoroughness In Paper Reading:**

I read the paper at least twice and used my best judgement in assessing the paper.

---

> ### Author Response · Authors · 2019-11-11
> **Response to Reviewer 4**
>
> We thank the reviewer for the comments and appreciation of our work. We definitely aim to evaluate larger Transformer models, but need to improve our codebase first. Our numeric simulator runs out of memory when running larger NMT datasets, and we are actively working on improvements. Regardless of acceptance, we are happy to update some results here when they become available.
>
> Thanks for the note on the master weights. We have updated the first section to clearly state that master weights and accumulations remain in FP32.
>
> We have also revised Section 2, Paragraphs 3 and 4 per your suggestion, and updated Equations (2) and (3) for readability and typos. Thanks for the suggestions.

---

### Official Review · AnonReviewer3 · 2019-11-06
**Official Blind Review #3**

**Rating:** 6

**Review:**

Good ideas
Important area
Impressive results.
Could be very useful for many embedded applications.

I’m not a hardware expert but can see why this would be Useful.



**Experience Assessment:**

I do not know much about this area.

**Review Assessment: Checking Correctness Of Derivations And Theory:**

N/A

**Review Assessment: Checking Correctness Of Experiments:**

I assessed the sensibility of the experiments.

**Review Assessment: Thoroughness In Paper Reading:**

I read the paper at least twice and used my best judgement in assessing the paper.

---

### Public Comment · ~Nimit_Sharad_Sohoni1 · 2019-11-08
**Comparison to "Scalable Methods for 8-bit Training of Neural Networks"?**

It would be interesting to see a comparison with the methods proposed in "Scalable Methods for 8-bit Training of Neural Networks" [NeurIPS '18, R. Banner, I. Hubara, E. Hoffer, D. Soudry]. (Also, this paper should probably be cited either way.)

---

> ### Author Response · Authors · 2019-11-11
> **Response to - Comparison to "Scalable Methods for 8-bit Training of Neural Networks"**
>
> We thank you for your suggestion.
>
> The Range Batch-Normalization idea mentioned in the paper normalizes the range of activations, per dimension to make Batch Normalization more robust to lower precision(8-bit) data. While there is an overlap in the underlying theme of both papers i.e. 8-bit training of networks, here are some of the key differences:
> 1. Our approach mainly focuses on eliminating the need for tuned loss-scaling and specific hardware rounding approaches like Stochastic Rounding. The above mentioned paper needed Stochastic Rounding for convergence as cited in Section 4 of their paper.
> 2. The above mentioned paper’s 8-bit quantization scheme is based on gemmlowp(https://arxiv.org/pdf/1712.05877.pdf) which leverages integer-arithmetic quantization showing convergence on ResNet-18/50. Our approach focuses on training with 8-bit floating point numbers instead proving convergence on wider range of representative models like ResNet, Transformer-tiny and Neural Collaborative Filtering.
> 3. In our approach, all gradients are in 8-bits(S2FP8) as compared to the above approach that maintains layer gradients in 16-bits and weight gradients in 8-bits.
>
> We have also updated the Related Work section of our paper citing the above reference.
> Further, we have updated the Appendix(Table A2) to compare S2FP8 with BF16 in addition to vanilla FP8, FP8 with tuned approaches already analyzed in the paper.

---

### Decision · Program_Chairs · 2019-12-19

**Decision:**

Accept (Poster)

**Comment:**

Main description:  paper focuses on training neural networks using 8-bit floating-point numbers (FP8). The goal is highly motivated: training neural networks faster, with smaller memory footprint and energy consumption.

Discussions
reviewer 3:  gives a very short review and is not knowledagble in this area (rating is weak accept)
reviewer 4: well written and convincing paper, some minor technical flaws (not very knowledgable)
reviewer 1: interesting paper but argues not very practical (not very knowledgable)
reviewer 2: this is the most thorough and knowledable review, and here the authors like the scope of the paper and its interest to ICLR.
Recommendation: going mainly by reviewer 2, i vote to accept this as a poster